# Self-pretraining for small datasets by exploiting patch information

## Abstract

Deep learning tasks with small datasets are often tackled by pretraining models with large datasets on relevent tasks. Although pretraining methods mitigate the problem of overfitting, it can be difficult to find appropriate pretrained models sometimes. In this paper, we proposed a self-pretraininng method by exploiting patch information in the dataset itself without pretraining on other datasets. Our experiments show that the self-pretraining method leads to better performance than training from scratch both in the condition of not using other data.

## 1 Introduction

Transfer learning has become the *de facto* approach of doing deep learning tasks on small datasets. Because of the data-hungry nature of deep learning methods, training from scratch using small datasets usually got overfitting. Although transfer learning using models pretrained on additional large datasets mitigates the problem of overfitting, it is hard to find an appropriate pretrained model like the ImageNet-classification pretrained model which used in detection and segmentation tasks when the appearance of input data or the goal of task in the target domain is special.

Research on training with small datasets without using external information has emerged in these years. Barz et al.(Barz & Denzler, 2020) proposed the method of training from scratch on small datasets using the cosine loss, which got substantially better performance than using the cross entropy loss function on fine-grained classification tasks. Zhang et al.(Zhang et al., 2019) introduced a generative adversarial network into the process of training with limited datasets without using external data or prior knowledge.

In contrast to doing data augmentation or training using special loss functions on small dataset tasks, we proposed the self-pretraining method which transfers patch information in the dataset itself to the model in a weakly supervised manner. Patches in images can represent image information in some extent. In (Kang et al., 2014), Kang et.al predicted the image quality score using the average quality score of image patches which are trained by image-level quality labels. Also, BagNet(Brendel & Bethge, 2019) indicated that small image patches which contain the class evidence can do well in the ImageNet classification challenge by aggregating the their score in the image without considering the spatial order. In a case of the fine-grained classification, (Wang et al., 2017) extracted features of image patches by training with external large datasets in a weakly supervised way. Inspired by (Gatys et al., 2016) which pointed out that convolution neural networks get local features in lower layers and global structure features in higher layers, our self-pretraining method pretrains lower layers to higher layers in the model using image pathces with the incremental size step by step.

## 2 Proposed method

In this section, we proposed our self-pretraining method using patch information in the dataset itself. Despite the small number of training images, the number of patches sampled from images can be large enough to meet the data-hungry demand.

Our self-pretraining methods get insights from two aspects:

First, large amounts of patches which contain parts of the information in the image can training the network using image-level labels in a weakly supervised manner. Although each small patch does

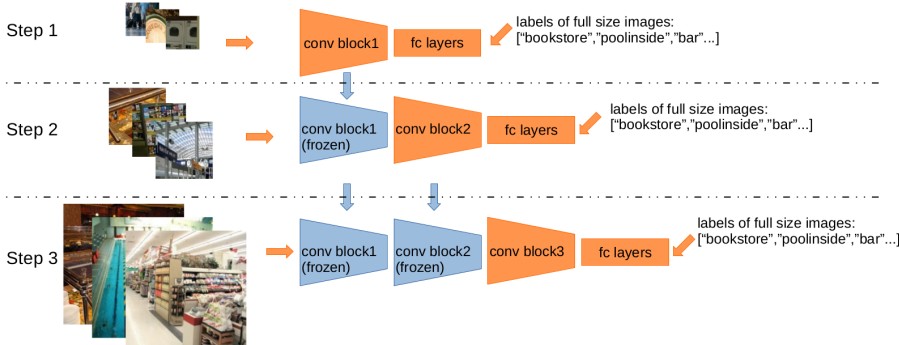

Figure 1: We get the prediction model with the input of full size images by pretraining with patches in a multi-step way. In the begining (Step 1), small patches are sampled as training data with image-level labels. Then (Step 2), the lower layers pretrained in the former step (Step 1) are frozen as the early feature extractor in the model of this step. Iteratively continue this procedure until (Step 3) the size of patches to be sampled is equal to the full size of images.

---

**Algorithm 1** Self-Pretraining Using Patch Information

---

**require:** dataset $\mathcal{D}$ with image of $r \times r$ size, incremental size list $r_0, r_1, ..., r_K$, incremental stride list $s_0, s_1, ..., s_K$, subject to $s_K = r_K = r$
$\mathcal{D}_0 \leftarrow$ sample patches of $r_0 \times r_0$ size with stride $s_0$ from $\mathcal{D}$ as dataset $\mathcal{D}_0$
$M_0 \leftarrow$ construct the model $M_0$ with the convolutional block $C_0$ and several fully connected layers
$M_0 \leftarrow$ Training $M_0$ with $\mathcal{D}_0$
**for** $k$ **in** $1, 2, ..., K$:
    $\mathcal{D}_k \leftarrow$ sample patches of $r_k \times r_k$ size with stride $s_k$ as dataset $\mathcal{D}_k$
    $M_k \leftarrow$ construct the model $M_k$ with sequential convolution blocks $C_0, C_1, ..., C_{k-1}$ with frozen weights followed by the convlutional block $C_k$ and several fully connected layers with randomly initialized weights
    $M_k \leftarrow$ Training $M_k$ with $\mathcal{D}_k$
**return** $M_K$

---

not hold the complete information leads to the correct prediction, the model trained on these small patches probably learned similar parameters in lower layers compared with models trained on large relevant datasets in a discriminative way. Therefore, traning using image patches with not exactly correct image-level labels may get the performance of the image feature extraction close to standard pretraining methods.

Otherwise, training with random shuffling image patches means that self-pretrained models do not use the global structure information to predict results, while special structures in limited data can cause the overfitting problem more easily than local patches which have more intra-class samples in the dataset.

The self-pretraining method pretrains the model using patches with incremental size in a step by step way, which can be seen in Figure 1.

In early stages of the self-pretraining, we sample smaller image patches uniformlly as training data which the number of them can be larger. The target of each patch is assigned with the label of the image containing this patch. After training within these patches in a weakly-supervised manner, we remove higher layers of the pretrained model while retain lower layers as the frozen feature extractor in the next training step, in which bigger patches are sampled from images as training data. The procedure continues iteratively until the size of patches to be sampled is equal to the full image size.

Using this step by step pretraining method, we can get a model with the input of full size images which has better performance than training from scratch on the small dataset. Detailed descriptions of self-pretraining methods are in Algorithm 1.

## 3 EXPERIMENTS

We test our self-pretraining method on two tasks: blind image quality assessment using LIVE dataset(Sheikh et al., 2006) and indoor scene classification using MIT Indoor dataset(Quattoni & Torralba, 2009). The datasets of these two tasks have small number of images, which would get worse results by the method of training from scratch.

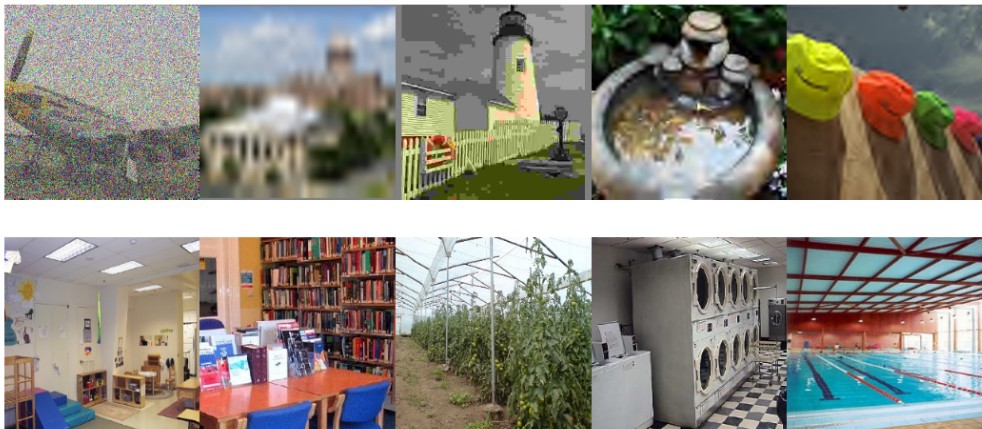

Figure 2: Samples of two datasets. Images in the top row are sampled from LIVE dataset while images in the bottom row are from MIT Indoor dataset.

### 3.1 BLIND IMAGE QUALITY ASSESSMENT (IQA)

For the IQA task, we use LIVE dataset, which have 779 distorted images with five different distortions — JP2k compression, JPEG compression, White Gaussian, Gaussian blur and Fast Fading at 7~8 degradation levels derived from 29 pristine reference images. Differential Mean Opinion Scores (DMOS) are provided for each image, roughly in the range [0, 100]. Higher DMOS indicates lower quality.

In this regression task, the problem of data scarcity is serious due to the small number of images in the dataset while getting worse beacuse of large variances in the appearance between images with similar labels. Images with totally different content can have similar quality scores (DMOS) while images with similar content can have largely different quality scores due to different levels of image degradation from a same reference image. To sovle to the problem above, the dataset should contain large amounts of images with various structures in similar quality score intervals.

Thanks to the property of this task in which the quality of images are distributed uniformly in each image space to some extent, the patches in the image give quite a lot information to the image quality target. We crop and pad the original image to $512 \times 512$, then sample patches of sizes with [32, 64, 128, 256, 512] from images step by step. As increasing the sample sizes, the frozen convolutional blocks are stacked to build the pretrained model from bottom to top. We train the model for 80 epochs using Adam optimizer with the learning rate of 0.001 and decrease it by 0.1 every 30 epochs, using the model in the final epoch as the final model.

With randomly splitting the LIVE datasets to 80% training dataset and 20% testing dataset, our method get a better result on the testing dataset than training from scratch. We get the image quality in the self-pretraining stage by averaging patch scores in the whole image. A gradually promotion in the performance can be seen in Table 1. Note that in the table, "Pearson" means Pearson Correlation Coefficient, "Spearman" means Spearman Correlation Coefficient, "MAE" means Mean Absolute Error and "scratch" means training from scratch using full size images.

Table 1: Results on LIVE dataset

| Input Size | Pearson | Spearman | MAE |
|---|---|---|---|
| 32 | 0.9649 | 0.9642 | 5.224 |
| 64 | 0.9675 | 0.9648 | 5.051 |
| 128 | 0.9704 | 0.9672 | 4.441 |
| 256 | **0.9748** | **0.9679** | 3.614 |
| 512 | 0.9743 | 0.9670 | **3.479** |
| 512(scratch) | 0.8910 | 0.8977 | 9.331 |

Table 2: Results on MIT Indoor67 dataset

| Input Size | Accuracy |
|---|---|
| 64 | 37.71% |
| 128 | 38.83% |
| 256 | **42.95%** |
| 256(scratch) | 39.88% |

## 3.2 INDOOR SCENE CLASSIFICATION

We experiment on the indoor scene classification task, in which patches in the images hold less information about the class evidence than the image quality assessment task above. We choose MIT Indoor67 dataset, which contains 67 indoor-scene classes and has 15,620 images in total. Each scene category contains at least 100 images, where 80 images are for training and 20 images for testing.

Patches in scene images can roughly give some information about the scene class. For example, patches with the content of books are more likely come from a bookstore image. We resize images to $256 \times 256$ sizes cause the image size makes no influence on this classification task, then sample uniformly with the size of [64, 128, 256] in our self-pretraining process. In the training procedure using $256 \times 256$ patches (full size images), we compare the self-pretraining method with the training from scratch method using the best test accuracy the model can get in the whole 80 training epochs due to the serious overfitting phenomenon which may makes it difficult to choose a best model if using splitted validation datasets.

We got gradually improving results as the size of patches increasing the the training process, seeing Table 2. Note that in this task, models trained on smaller patches are worse than the model training from scratch using full size images, until the final self-pretraining model with $256 \times 256$ full size inputs get better result than it.

## 4 CONCLUSION

In this paper, we proposed the self-pretraining method using the dataset itself, which boosts the performance of the pretrained model step by step using information of patches. Although problems can be solved by collecting more and more data in the big-data era, self-pretraining methods can sometimes benefit specific tasks with the data which the number is limited or with too much image structure variances within the class. We hope our work can also motivate the research on fully exploiting the information in the dataset itself according to the specific property of learning tasks.

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
