# OpenReview forum: "Self-Pretraining for Small Datasets by Exploiting Patch Information"
_ICLR.cc/2021/Conference — Reject_

### Official Review · AnonReviewer1 · 2020-10-25
**Good idea**

**Rating:** 4
**Confidence:** 4

**Review:**

This paper presents an interesting approach for training neural networks with a small dataset. The main idea is to train the model from the early layers to the deeper layers step-by-step, with different types of inputs (i.e., patches, cropped images, or full images) sampled from the given training set. Experimental results show great performance compared to training from scratch.

Pros:
1. This paper presents a good idea for training with small dataset. The proposed method is technical valid, and it is clear that step-wise training can help improve the performance. From my point of view, it is more like applying intermediate supervision on each layer using different types of training inputs, By doing so, we ensure that the early layers and the deeper layers are forced to learn to find the specified low-level and high-level semantics, respectively. Thus, the resultant model could behaves like the large-scale pre-trained deep CNNs we analyzed in the literature.
2. In experiment section, the authors present the key results on two datasets showing the advantage of the proposed method.
3. The paper is easy to understand. Also, the writing is very concise.

Cons:
1. Though this paper presents a really focused contribution on training with small dataset, one can see that the paper lacks of in-depth analysis on either the target task or the proposed algorithm. I would suggest that the authors could conduct more experiments to better validate the target task (i.e., training with small dataset). It would be great to add a transfer learning baseline (i.e., pre-trained on ImageNet, and then fine-tuned on the target dataset), and show that it does not work for your research problem. The readers could better understand the difficulty of your research problem.
2. My another question is more related to the problem definition, or more specifically the importance of the addressed problem. (1) Why we need to deal with small datasets? If the target problem/application is important, it should be easy to enlarge the dataset at scale. For example, there is a scene classification dataset built by MIT in 2009. It is a small-scale dataset which is used in this paper. In 2015, due to the importance of the task, MIT people have scaled the dataset and the new one is called Places, which has 2.5 millions of images with scene-category labels. (2) Why it is small? There are two possible reasons I could think of: (i) it may be difficult and expensive to collect training labels due to extreme labor efforts or privacy concerns (i.e., pixel-wise labels, medical images). (ii) the applications are newly emerged so all data are sparse but surely it will be scaled up in the nearly future. But in the paper, the presented tasks, including scene classification and quality assessment, are well-established and they should not be so difficult to obtain training data. In general, I think the presented experiments are toy examples and the small-scale setting may not be convincing enough. I would suggest that the authors could include more examples and results of small-dataset scenarios, which could add values to the paper.

##################################
Post-rebuttal:
The idea is good, but the experiments and analysis are not enough to validate the proposed idea. The paper is not ready for a publication.

---

### Official Review · AnonReviewer3 · 2020-10-28
**It seems to be a draft of the idea rather than a ready paper**

**Rating:** 2
**Confidence:** 5

**Review:**

The paper presents a method for improving the accuracy when training CNNs from scratch on a small dataset.

The method is as following: instead of training the whole network on the full images, one first creates a shallow model and trains it on the small crops from the dataset. The labels for the patches are set the same, as for the whole image. Then one adds more layers, freezes the previously trained and train on bigger patches and so on.

The method is evaluated on 2 datasets: image quality assesment on LIVE dataset and scene classification on MIT Indoor 67.


******

While the idea might be good, the paper itself is a low quality.
Problems:


1. Two things are mixed together: (a) progressive patch cropping, starting from small and ending up with big image and (b) progressive adding more and more layers, while freezeing earliest.

That is not clear and justified to use both together.

The cropping can be seen is very aggressize data augmentation and if fact is commonly used for training ImageNet CNNs, e.g. see [2].

The freezing and training is a variant of Net2Net[1], which is not cited.


2. Datasets useBlind image quality assessment is a bad task for testing the idea. Why? Because when image is of a bad quality, it can often easily be told from a small patch. That is why aggressize patch cropping and  keeping label the same is justified. That is not true for other tasks, like classification, object detection, metric learning and so on. Yet the method is claimed to be quite general.

3. There is no baseline on any of the standard dataset, not strong supervised baseline on the datasets paper proposed. By strong I mean, where hyperparameters are reasonably tuned and standard data augmentation is used.
E.g. for the MIT Indoor67 the paper reports 42.95% accuracy. In the same time, course report from 2015 [3] has 43.8% accuracy. I don't see improvement from the proposed method.

4. The model is not even specified. Is it custom CNN? VGG-style? ResNet-style?


***

I would recommend to start from the strong baseline and proper evaluation. If the method can improve on them and then  well presented, then it can be published.


[1]. Net2Net: Accelerating Learning via Knowledge Transfer. Chen et.al, ICLR 2016 https://arxiv.org/abs/1511.05641

[2]. https://github.com/NVIDIA/DALI/blob/1e9196702d991d3342ad7a5a7d57c2893abad832/docs/examples/use_cases/pytorch/resnet50/main.py#L116

[3]. http://cs231n.stanford.edu/reports/2015/pdfs/ondieki_final_paper.pdf

***

## After rebuttal update.

Given there is no rebuttal, there will be no update.

---

### Official Review · AnonReviewer2 · 2020-10-30
**Interesting problem, simple idea, poor experimental evaluation**

**Rating:** 4
**Confidence:** 5

**Review:**

The authors propose an interesting problem where only a small sized labeled data set is available for supervised learning. This approach is different from the existing deep learning benchmarks which either assume the availability of pre-trained supervised imageNet classification model or an unsupervised model trained using large scale auxiliary data (e.g. MoCo, Swav). The proposed method divides training of different stages of the convNet architecture using different sized crops/patches of images. The lower layers are trained to classify the smallest size crops. These layers are then frozen to train higher level layers with larger patches as inputs. The network learns to classify input patches into the classes of images they belong to. Evaluation is performed on image quality assessment task on Live dataset and indoor scene classification on MIT Indoor Scenes dataset.

The problem is of practical interest since a large amount of labeled/unlabeled data is not always available for the domain of interest (e.g. depth data etc). However, I am not entirely sure that methods using small scale data alone can ever achieve performances close to the methods that leverage pre-trained supervised/unsupervised prior models. Even in cases where large scale data is unavailable, small scale paired data (e.g. image + depth ) can be used to regularize the training of one domain using a pre-trained model of the other domain [e.g. Gupta et. al. CVPR 2016].
The experimental evaluation in this paper is woefully inadequate. It is clear that the method is more successful compared to learning from scratch, but I would have liked to see how much worse it is compared to fine-tuning of pre-trained models. If the gap is too large, is it even a direction worth exploring? Is the proposed method, the first attempt at training neural nets without priors? If not, how does it compare to others? Evaluation on more serious datasets such as SUN 397 would have been more appealing. At present it seems like a simple experiment on a couple small datasets.

---

### Decision · Program_Chairs · 2021-01-07
**Final Decision**

**Decision:**

Reject

**Comment:**

All reviewers agreed on the major shortcomings of this submission, the most important of which is that the contributions are insufficiently evaluated. There was no author response.